# Deep Equals Shallow for ReLU Networks in Kernel Regimes

**Alberto Bietti**[*]
NYU[†]
alberto.bietti@nyu.edu

**Francis Bach**
Inria[‡]
francis.bach@inria.fr

## Abstract

Deep networks are often considered to be more expressive than shallow ones in terms of approximation. Indeed, certain functions can be approximated by deep networks provably more efficiently than by shallow ones, however, no tractable algorithms are known for learning such deep models. Separately, a recent line of work has shown that deep networks trained with gradient descent may behave like (tractable) kernel methods in a certain over-parameterized regime, where the kernel is determined by the architecture and initialization, and this paper focuses on approximation for such kernels. We show that for ReLU activations, the kernels derived from deep fully-connected networks have essentially the same approximation properties as their "shallow" two-layer counterpart, namely the same eigenvalue decay for the corresponding integral operator. This highlights the limitations of the kernel framework for understanding the benefits of such deep architectures. Our main theoretical result relies on characterizing such eigenvalue decays through differentiability properties of the kernel function, which also easily applies to the study of other kernels defined on the sphere.

## 1 Introduction

The question of which functions can be well approximated by neural networks is crucial for understanding when these models are successful, and has always been at the heart of the theoretical study of neural networks (*e.g.*, Hornik et al., 1989; Pinkus, 1999). While early works have mostly focused on shallow networks with only two layers, more recent works have shown benefits of deep networks for approximating certain classes of functions (Eldan & Shamir, 2016; Mhaskar & Poggio, 2016; Telgarsky, 2016; Daniely, 2017; Yarotsky, 2017; Schmidt-Hieber et al., 2020). Unfortunately, many of these approaches rely on constructions that are not currently known to be learnable using efficient algorithms.

A separate line of work has considered over-parameterized networks with random neurons (Neal, 1996), which also display universal approximation properties while additionally providing efficient algorithms based on kernel methods or their approximations such as random features (Rahimi & Recht, 2007; Bach, 2017b). Many recent results on gradient-based optimization of certain over-parameterized networks have been shown to be equivalent to kernel methods with an architecture-specific kernel called the *neural tangent kernel* (NTK) and thus also fall in this category (*e.g.*, Jacot et al., 2018; Li & Liang, 2018; Allen-Zhu et al., 2019b; Du et al., 2019a;b; Zou et al., 2019). This regime has been coined *lazy* (Chizat et al., 2019), as it does not capture the common phenomenon where weights move significantly away from random initialization and thus may not provide a satisfying model for learning adaptive representations, in contrast to other settings such as the *mean field* or *active* regime, which captures complex training dynamics where weights may move in a non-trivial manner and adapt to the data (*e.g.*, Chizat & Bach, 2018; Mei et al., 2018). Nevertheless,

---

[*]Work done while at Inria.

[†]Center for Data Science, New York University. New York, USA.

[‡]Inria - Département d'Informatique de l'École Normale Supérieure. PSL Research University. Paris, France.

one benefit compared to the mean field regime is that the kernel approach easily extends to deep architectures, leading to compositional kernels similar to the ones of Cho & Saul (2009); Daniely et al. (2016). Our goal in this paper is to study the role of depth in determining approximation properties for such kernels, with a focus on fully-connected deep ReLU networks.

Our approximation results rely on the study of eigenvalue decays of integral operators associated to the obtained dot-product kernels on the sphere, which are diagonalized in the basis of spherical harmonics. This provides a characterization of the functions in the corresponding reproducing kernel Hilbert space (RKHS) in terms of their smoothness, and leads to convergence rates for non-parametric regression when the data are uniformly distributed on the sphere. We show that for ReLU networks, the eigenvalue decays for the corresponding deep kernels remain the same regardless of the depth of the network. Our key result is that the decay for a certain class of kernels is characterized by a property related to differentiability of the kernel function around the point where the two inputs are aligned. In particular, the property is preserved when adding layers with ReLU activations, showing that depth plays essentially no role for such networks in kernel regimes. This highlights the limitations of the kernel regime for understanding the power of depth in fully-connected networks, and calls for new models of deep networks beyond kernels (see, *e.g.*, Allen-Zhu & Li, 2020; Chen et al., 2020, for recent works in this direction). We also provide applications of our result to other kernels and architectures, and illustrate our results with numerical experiments on synthetic and real datasets.

**Related work.** Kernels for deep learning were originally derived by Neal (1996) for shallow networks, and later for deep networks (Cho & Saul, 2009; Daniely et al., 2016; Lee et al., 2018; Matthews et al., 2018). Smola et al. (2001); Minh et al. (2006) study regularization properties of dot-product kernels on the sphere using spherical harmonics, and Bach (2017a) derives eigenvalue decays for such dot-product kernels arising from shallow networks with positively homogeneous activations including the ReLU. Extensions to shallow NTK or Laplace kernels are studied by Basri et al. (2019); Bietti & Mairal (2019b); Geifman et al. (2020). The observation that depth does not change the decay of the NTK was previously made by Basri et al. (2020) empirically, and Geifman et al. (2020) provide a lower bound on the eigenvalues for deep networks; our work makes this observation rigorous by providing tight asymptotic decays. Spectral properties of wide neural networks were also considered in (Cao et al., 2019; Fan & Wang, 2020; Ghorbani et al., 2019; Xie et al., 2017; Yang & Salman, 2019). Azevedo & Menegatto (2014); Scetbon & Harchaoui (2020) also study eigenvalue decays for dot-product kernels but focus on kernels with geometric decays, while our main focus is on polynomial decays. Additional works on over-parameterized or infinite-width networks in lazy regimes include (Allen-Zhu et al., 2019a;b; Arora et al., 2019a;b; Brand et al., 2020; Lee et al., 2020; Song & Yang, 2019).

Concurrently to our work, Chen & Xu (2021) also studied the RKHS of the NTK for deep ReLU networks, showing that it is the same as for the Laplace kernel on the sphere. They achieve this by studying asymptotic decays of Taylor coefficients of the kernel function at zero using complex-analytic extensions of the kernel functions, and leveraging this to obtain both inclusions between the two RKHSs. In contrast, we obtain precise descriptions of the RKHS and regularization properties in the basis of spherical harmonics for various dot-product kernels through spectral decompositions of integral operators, using (real) asymptotic expansions of the kernel function around endpoints. The equality between the RKHS of the deep NTK and Laplace kernel then easily follows from our results by the fact that the two kernels have the same spectral decay.

## 2    REVIEW OF APPROXIMATION WITH DOT-PRODUCT KERNELS

In this section, we provide a brief review of the kernels that arise from neural networks and their approximation properties.

## 2.1 Kernels for wide neural networks

Wide neural networks with random weights or weights close to random initialization naturally lead to certain dot-product kernels that depend on the architecture and activation function, which we now present, with a focus on fully-connected architectures.

**Random feature kernels.** We first consider a two-layer (shallow) network of the form $f(x) = \frac{1}{\sqrt{m}} \sum_{j=1}^m v_j \sigma(w_j^\top x)$, for some activation function $\sigma$. When $w_j \sim \mathcal{N}(0, I) \in \mathbb{R}^d$ are fixed and only $v_j \in \mathbb{R}$ are trained with $\ell_2$ regularization, this corresponds to using a random feature approximation Rahimi & Recht (2007) of the kernel

$$k(x, x') = \mathbb{E}_{w \sim \mathcal{N}(0,I)}[\sigma(w^\top x)\sigma(w^\top x')]. \tag{1}$$

If $x, x'$ are on the sphere, then by spherical symmetry of the Gaussian distribution, one may show that $k$ is invariant to unitary transformations and takes the form $k(x, x') = \kappa(x^\top x')$ for a certain function $\kappa$. More precisely, if $\sigma(u) = \sum_{i \geq 0} a_i h_i(u)$ is the decomposition of $\sigma$ in the basis of Hermite polynomials $h_i$, which are orthogonal w.r.t. the Gaussian measure, then we have (Daniely et al., 2016):

$$\kappa(u) = \sum_{i \geq 0} a_i^2 u^i. \tag{2}$$

Conversely, given a kernel function of the form above with $\kappa(u) = \sum_{i \geq 0} b_i u^i$ with $b_i \geq 0$, one may construct corresponding activations using Hermite polynomials by taking

$$\sigma(u) = \sum_i a_i h_i(u), \quad a_i \in \{\pm\sqrt{b_i}\}. \tag{3}$$

In the case where $\sigma$ is $s$-positively homogeneous, such as the ReLU $\sigma(u) = \max(u, 0)$ (with $s = 1$), or more generally $\sigma_s(u) = \max(u, 0)^s$, then the kernel (1) takes the form $k(x, x') = \|x\|^s \|x'\|^s \kappa(\frac{x^\top x'}{\|x\|\|x'\|})$ for any $x, x'$. This leads to RKHS functions of the form $f(x) = \|x\|^s g(\frac{x}{\|x\|})$, with $g$ in the RKHS of the kernel restricted to the sphere (Bietti & Mairal, 2019b, Prop. 8). In particular, for the step and ReLU activations $\sigma_0$ and $\sigma_1$, the functions $\kappa$ are given by the following arc-cosine kernels (Cho & Saul, 2009):[1]

$$\kappa_0(u) = \frac{1}{\pi}(\pi - \arccos(u)), \qquad \kappa_1(u) = \frac{1}{\pi}\left(u \cdot (\pi - \arccos(u)) + \sqrt{1 - u^2}\right). \tag{4}$$

Note that given a kernel function $\kappa$, the corresponding activations (3) will generally not be homogeneous, thus the inputs to a random network with such activations need to lie on the sphere (or be appropriately normalized) in order to yield the kernel $\kappa$.

**Extension to deep networks.** When considering a deep network with more than two layers and fixed random weights before the last layer, the connection to random features is less direct since the features are correlated through intermediate layers. Nevertheless, when the hidden layers are wide enough, one still approaches a kernel obtained by letting the widths go to infinity (see, *e.g.*, Daniely et al., 2016; Lee et al., 2018; Matthews et al., 2018), which takes a similar form to the multi-layer kernels of Cho & Saul (2009):

$$k^L(x, x') = \kappa^L(x^\top x') := \underbrace{\kappa \circ \cdots \circ \kappa}_{L-1 \text{ times}}(x^\top x'),$$

for $x, x'$ on the sphere, where $\kappa$ is obtained as described above for a given activation $\sigma$, and $L$ is the number of layers. We still refer to this kernel as the *random features* (RF) kernel in this paper, noting that it is sometimes known as the "conjugate kernel" or NNGP kernel (for neural network Gaussian process). It is usually good to normalize $\kappa$ such that $\kappa(1) = 1$, so that we also have $\kappa^L(1) = 1$, avoiding exploding or vanishing behavior for deep networks. In practice, this corresponds to using an activation-dependent scaling in the random weight initialization, which is commonly used by practitioners (He et al., 2015).

---

[1] Here we assume a scaling $\sqrt{2/m}$ instead of $\sqrt{1/m}$ in the definition of $f$, which yields $\kappa(1) = 1$, a useful normalization for deep networks, as explained below.

**Neural tangent kernels.** When intermediate layers are trained along with the last layer using gradient methods, the resulting problem is non-convex and the statistical properties of such approaches are not well understood in general, particularly for deep networks. However, in a specific over-parameterized regime, it may be shown that gradient descent can reach a global minimum while keeping weights very close to random initialization. More precisely, for a network $f(x; \theta)$ parameterized by $\theta$ with large width $m$, the model remains close to its linearization around random initialization $\theta_0$ throughout training, that is, $f(x; \theta) \approx f(x; \theta_0) + \langle \theta - \theta_0, \nabla_\theta f(x; \theta_0) \rangle$. This is also known as the *lazy training* regime (Chizat et al., 2019). Learning is then equivalent to a kernel method with another architecture-specific kernel known as the *neural tangent kernel* (NTK, Jacot et al., 2018), given by

$$k_{\mathrm{NTK}}(x, x') = \lim_{m \to \infty} \langle \nabla f(x; \theta_0), \nabla f(x'; \theta_0) \rangle. \tag{5}$$

For a simple two-layer network with activation $\sigma$, it is then given by

$$k_{\mathrm{NTK}}(x, x') = (x^\top x') \ \mathbb{E}_w[\sigma'(w^\top x)\sigma'(w^\top x')] + \mathbb{E}_w[\sigma(w^\top x)\sigma(w^\top x')]. \tag{6}$$

For a ReLU network with $L$ layers with inputs on the sphere, taking appropriate limits on the widths, one can show (Jacot et al., 2018): $k_{\mathrm{NTK}}(x, x') = \kappa_{\mathrm{NTK}}^L(x^\top x')$, with $\kappa_{\mathrm{NTK}}^1(u) = \kappa^1(u) = u$ and for $\ell = 2, \ldots, L$,

$$\kappa^\ell(u) = \kappa_1(\kappa^{\ell-1}(u))$$
$$\kappa_{\mathrm{NTK}}^\ell(u) = \kappa_{\mathrm{NTK}}^{\ell-1}(u)\kappa_0(\kappa^{\ell-1}(u)) + \kappa^\ell(u), \tag{7}$$

where $\kappa_0$ and $\kappa_1$ are given in (4).

## 2.2 Approximation and harmonic analysis with dot-product kernels

In this section, we recall approximation properties of dot-product kernels on the sphere, through spectral decompositions of integral operators in the basis of spherical harmonics. Further background is provided in Appendix A.

**Spherical harmonics and description of the RKHS.** A standard approach to study the RKHS of a kernel is through the spectral decomposition of an integral operator $T$ given by $Tf(x) = \int k(x, y)f(y)d\tau(y)$ for some measure $\tau$, leading to Mercer's theorem (*e.g.*, Cucker & Smale, 2002). When inputs lie on the sphere $\mathbb{S}^{d-1}$ in $d$ dimensions, dot-product kernels of the form $k(x, x') = \kappa(x^\top x')$ are rotationally-invariant, depending only on the angle between $x$ and $x'$. Similarly to how translation-invariant kernels are diagonalized in the Fourier basis, rotation-invariant kernels are diagonalized in the basis of spherical harmonics (Smola et al., 2001; Bach, 2017a), which lead to connections between eigenvalue decays and regularity as in the Fourier setting. In particular, if $\tau$ denotes the uniform measure on $\mathbb{S}^{d-1}$, then $TY_{k,j} = \mu_k Y_{k,j}$, where $Y_{k,j}$ is the $j$-th spherical harmonic polynomial of degree $k$, where $k$ plays the role of a frequency as in the Fourier case, and the number of such orthogonal polynomials of degree $k$ is given by $N(d, k) = \frac{2k+d-2}{k}\binom{k+d-3}{d-2}$, which grows as $k^{d-2}$ for large $k$. The eigenvalues $\mu_k$ only depend on the frequency $k$ and are given by

$$\mu_k = \frac{\omega_{d-2}}{\omega_{d-1}} \int_{-1}^{1} \kappa(t)P_k(t)(1-t^2)^{(d-3)/2}dt, \tag{8}$$

where $P_k$ is the Legendre polynomial of degree $k$ in $d$ dimensions (also known as Gegenbauer polynomial when using a different scaling), and $\omega_{d-1}$ denotes the surface of the sphere $\mathbb{S}^{d-1}$. Mercer's theorem then states that the RKHS $\mathcal{H}$ associated to the kernel is given by

$$\mathcal{H} = \left\{ f = \sum_{k \geq 0, \mu_k \neq 0} \sum_{j=1}^{N(d,k)} a_{k,j} Y_{k,j}(\cdot) \quad \text{s.t.} \quad \|f\|_{\mathcal{H}}^2 := \sum_{k \geq 0, \mu_k \neq 0} \sum_{j=1}^{N(d,k)} \frac{a_{k,j}^2}{\mu_k} < \infty \right\}. \tag{9}$$

In particular, if $\mu_k$ has a fast decay, then the coefficients $a_{k,j}$ of $f$ must also decay quickly with $k$ in order for $f$ to be in $\mathcal{H}$, which means $f$ must have a certain level of regularity. Similarly to the Fourier case, an exponential decay of $\mu_k$ implies that the functions in $\mathcal{H}$ are

infinitely differentiable, while for polynomial decay $\mathcal{H}$ contains all functions whose derivatives only up to a certain order are bounded, as in Sobolev spaces. If two kernels lead to the same asymptotic decay of $\mu_k$ up to a constant, then by (9) their RKHS norms are equivalent up to a constant, and thus they have the same RKHS. For the specific case of random feature kernels arising from $s$-positively homogeneous activations, Bach (2017a) shows that $\mu_k$ decays as $k^{-d-2s}$ for $k$ of the opposite parity of $s$, and is zero for large enough $k$ of opposite parity, which results in a RKHS that contains even or odd functions (depending on the parity of $s$) defined on the sphere with bounded derivatives up to order $\beta := d/2 + s$ (note that $\beta$ must be greater than $(d-1)/2$ in order for the eigenvalues of $T$ to be summable and thus lead to a well-defined RKHS). Bietti & Mairal (2019b) show that the same decay holds for the NTK of two-layer ReLU networks, with $s = 0$ and a change of parity. Basri et al. (2019) show that the parity constraints may be removed by adding a zero-initialized additive bias term when deriving the NTK. We note that one can also obtain rates of approximation for Lipschitz functions from such decay estimates (Bach, 2017a). Our goal in this paper is to extend this to more general dot-product kernels such as those arising from multi-layer networks, by providing a more general approach for obtaining decay estimates from differentiability properties of the function $\kappa$.

**Non-parametric regression.** When the data are uniformly distributed on the sphere, we may also obtain convergence rates for non-parametric regression, which typically depend on the eigenvalue decay of the integral operator associated to the marginal distribution on inputs and on the decomposition of the regression function $f^*(x) = \mathbb{E}[y|x]$ on the same basis (*e.g.*, Caponnetto & De Vito, 2007).[2] Then one may achieve optimal rates that depend mainly on the regularity of $f^*$ when using various algorithms with tuned hyperparameters, but the choice of kernel and its decay may have an impact on the rates in some regimes, as well as on the difficulty of the optimization problem (see, *e.g.*, Bach, 2013, Section 4.3).

# 3 Main Result and Applications to Deep Networks

In this section, we present our main results concerning approximation properties of dot-product kernels on the sphere, and applications to the kernels arising from wide random neural networks. We begin by stating our main theorem, which provides eigenvalue decays for dot-product kernels from differentiability properties of the kernel function $\kappa$ at the end-points $\pm 1$. We then present applications of this result to various kernels, including those coming from deep networks, showing in particular that the RKHSs associated to deep and shallow ReLU networks are the same (up to parity constraints).

## 3.1 Statement of our main theorem

We now state our main result regarding the asymptotic eigenvalue decay of dot-product kernels. Recall that we consider a kernel of the form $k(x, y) = \kappa(x^\top y)$ for $x, y \in \mathbb{S}^{d-1}$, and seek to obtain decay estimates on the eigenvalues $\mu_k$ defined in (8). We now state our main theorem, which derives the asymptotic decay of $\mu_k$ with $k$ in terms of differentiability properties of $\kappa$ around $\{\pm 1\}$, assuming that $\kappa$ is infinitely differentiable on $(-1, 1)$. This latter condition is always verified when $\kappa$ takes the form of a power series (2) with $\kappa(1) = 1$, since the radius of convergence is at least 1. We also require a technical condition, namely the ability to "differentiate asymptotic expansions" of $\kappa$ at $\pm 1$, which holds for the kernels considered in this work.

**Theorem 1** (Decay from regularity of $\kappa$ at endpoints, simplified). *Let $\kappa : [-1, 1] \to \mathbb{R}$ be a function that is $C^\infty$ on $(-1, 1)$ and has the following asymptotic expansions around $\pm 1$:*

$$\kappa(1 - t) = p_1(t) + c_1 t^\nu + o(t^\nu) \tag{10}$$

$$\kappa(-1 + t) = p_{-1}(t) + c_{-1} t^\nu + o(t^\nu), \tag{11}$$

*for $t \geq 0$, where $p_1, p_{-1}$ are polynomials and $\nu > 0$ is not an integer. Also, assume that the derivatives of $\kappa$ admit similar expansions obtained by differentiating the above ones. Then, there is an absolute constant $C(d, \nu)$ depending on $d$ and $\nu$ such that:*

---

[2]The rates easily extend to distributions with a density w.r.t. the uniform distribution on the sphere, although the eigenbasis on which regularity is measured is then different.

- *For $k$ even, if $c_1 \neq -c_{-1}$: $\mu_k \sim (c_1 + c_{-1})C(d, \nu)k^{-d-2\nu+1}$;*
- *For $k$ odd, if $c_1 \neq c_{-1}$: $\mu_k \sim (c_1 - c_{-1})C(d, \nu)k^{-d-2\nu+1}$.*

*In the case $|c_1| = |c_{-1}|$, then we have $\mu_k = o(k^{-d-2\nu+1})$ for one of the two parities (or both if $c_1 = c_{-1} = 0$). If $\kappa$ is infinitely differentiable on $[-1, 1]$ so that no such $\nu$ exists, then $\mu_k$ decays faster than any polynomial.*

The full theorem is given in Appendix B along with its proof, and requires an additional mild technical condition on the expansion which is verified for all kernels considered in this paper, namely, a finite number of terms in the expansions with exponents between $\nu$ and $\nu + 1$. The proof relies on integration by parts using properties of Legendre polynomials, in a way reminiscent of fast decays of Fourier series for differentiable functions, and on precise computations of the decay for simple functions of the form $t \mapsto (1 - t^2)^\nu$. This allows us to obtain the asymptotic decay for general kernel functions $\kappa$ as long as the behavior around the endpoints is known, in contrast to previous approaches which rely on the precise form of $\kappa$, or of the corresponding activation in the case of arc-cosine kernels (Bach, 2017a; Basri et al., 2019; Bietti & Mairal, 2019b; Geifman et al., 2020). This enables the study of more general and complex kernels, such as those arising from deep networks, as discussed below. When $\kappa$ is of the form $\kappa(t) = \sum_k b_k t^k$, the exponent $\nu$ in Theorem 1 is also related to the decay of coefficients $b_k$. Such coefficients provide a dimension-free description of the kernel which may be useful for instance in the study of kernel methods in certain high-dimensional regimes (see, *e.g.*, El Karoui, 2010; Ghorbani et al., 2019; Liang et al., 2020). We show in Appendix B.1 that the $b_k$ may be recovered from the $\mu_k$ by taking high-dimensional limits $d \to \infty$, and that they decay as $k^{-\nu-1}$.

## 3.2 Consequences for ReLU networks

When considering neural networks with ReLU activations, the corresponding random features and neural tangent kernels depend on the arc-cosine functions $\kappa_1$ and $\kappa_0$ defined in (4). These have the following expansions (with generalized exponents) near $+1$:

$$\kappa_0(1 - t) = 1 - \frac{\sqrt{2}}{\pi}t^{1/2} + O(t^{3/2}) \tag{12}$$

$$\kappa_1(1 - t) = 1 - t + \frac{2\sqrt{2}}{3\pi}t^{3/2} + O(t^{5/2}). \tag{13}$$

Indeed, the first follows from integrating the expansion of the derivative using the relation $\frac{d}{dt}\arccos(1 - t) = \frac{1}{\sqrt{2t}\sqrt{1-t/2}}$ and the second follows from the first using the expression of $\kappa_1$ in (4). Near $-1$, we have by symmetry $\kappa_0(-1 + t) = 1 - \kappa_0(1 - t) = \frac{\sqrt{2}}{\pi}t^{1/2} + O(t^{3/2})$, and we have $\kappa_1(-1 + t) = \frac{2\sqrt{2}}{3\pi}t^{3/2} + O(t^{5/3})$ by using $\kappa_1' = \kappa_0$ and $\kappa_1(-1) = 0$. The ability to differentiate the expansions follows from (Flajolet & Sedgewick, 2009, Theorem VI.8, p.419), together with a complex-analytic property known as $\Delta$-analyticity, which was shown to hold for RF and NTK kernels by Chen & Xu (2021). By Theorem 1, we immediately obtain a decay of $k^{-d-2}$ for even coefficients for $\kappa_1$, and $k^{-d}$ for odd coefficients for $\kappa_0$, recovering results of Bach (2017a). For the two-layer ReLU NTK, we have $\kappa_{\mathrm{NTK}}^2(u) = u\kappa_0(u) + \kappa_1(u)$, leading to a similar expansion to $\kappa_0$ and thus decay, up to a change of parity due to the factor $u$ which changes signs in the expansion around $-1$; this recovers Bietti & Mairal (2019b). We note that for these specific kernels, Bach (2017a); Bietti & Mairal (2019b) show in addition that coefficients of the opposite parity are exactly zero for large enough $k$, which imposes parity constraints on functions in the RKHS, although such a constraint may be removed in the NTK case by adding a zero-initialized bias term (Basri et al., 2019), leading to a kernel $\kappa_{\mathrm{NTK},b}(u) = (u + 1)\kappa_0(u) + \kappa_1(u)$.

**Deep networks.** Recall from Section 2.1 that the RF and NTK kernels for deep ReLU networks may be obtained through compositions and products using the functions $\kappa_1$ and $\kappa_0$. Since asymptotic expansions can be composed and multiplied, we can then obtain expansions for the deep RF and NTK kernels. The following results show that such kernels have the same eigenvalue decay as the ones for the corresponding shallow (two-layer) networks.

**Corollary 2** (Deep RF decay.)**.** *For the random neuron kernel $\kappa_{RF}^L$ of an L-layer ReLU network with $L \geq 3$, we have $\mu_k \sim C(d, L)k^{-d-2}$, where $C(d, L)$ is different depending on the parity of $k$ and grows linearly with $L$.*

**Corollary 3** (Deep NTK decay.)**.** *For the neural tangent kernel $\kappa_{NTK}^L$ of an L-layer ReLU network with $L \geq 3$, we have $\mu_k \sim C(d, L)k^{-d}$, where $C(d, L)$ is different depending on the parity of $k$ and grows quadratically with $L$ (it grows linearly with $L$ when considering the normalized NTK $\kappa_{NTK}^L/L$, which satisfies $\kappa_{NTK}^L(1)/L = 1$).*

The proofs, given in Appendix C, use the fact that $\kappa_1 \circ \kappa_1$ and $\kappa_1$ have the same non-integer exponent factors in their expansions, and similarly for $\kappa_0 \circ \kappa_1$ and $\kappa_0$. One benefit compared to the shallow case is that the odd and even coefficients are both non-zero with the same decay, which removes the parity constraints, but as mentioned before, simple modifications of the shallow kernels can yield the same effect.

**The finite neuron case.** For two-layer networks with a finite number of neurons, the obtained models correspond to random feature approximations of the limiting kernels (Rahimi & Recht, 2007). Then, one may approximate RKHS functions and achieve optimal rates in non-parametric regression as long as the number of random features exceeds a certain degrees-of-freedom quantity (Bach, 2017b; Rudi & Rosasco, 2017), which is similar to standard such quantities in the analysis of ridge regression (Caponnetto & De Vito, 2007), at least when the data are uniformly distributed on the sphere (otherwise the quantity involved may be larger unless features are sampled non-uniformly). Such a number of random features is optimal for a given eigenvalue decay of the integral operator (Bach, 2017b), which implies that the shallow random feature architectures provides optimal approximation for the multi-layer ReLU kernels as well, since the shallow and deep kernels have the same decay, up to the parity constraint. In order to overcome this constraint for shallow kernels while preserving decay, one may consider vector-valued random features of the form $(\sigma(w^\top x), x_1\sigma(w^\top x), \ldots, x_d\sigma(w^\top x))$ with $w \sim \mathcal{N}(0, I)$, leading to a kernel $\kappa_{\sigma,b}(u) = (1 + u)\kappa_\sigma(u)$, where $\kappa_\sigma$ is the random feature kernel corresponding to $\sigma$. With $\sigma(u) = \max(0, u)$, $\kappa_{\sigma,b}$ has the same decay as $\kappa_{RF}^L$, and when $\sigma(u) = \mathbb{1}\{u \geq 0\}$ it has the same decay as $\kappa_{NTK}^L$.

## 3.3 EXTENSIONS TO OTHER KERNELS

We now provide other examples of kernels for which Theorem 1 provides approximation properties thanks to its generality.

**Laplace kernel and generalizations.** The Laplace kernel $k_c(x, y) = e^{-c\|x-y\|}$ has been found to provide similar empirical behavior to neural networks when fitting randomly labeled data with gradient descent (Belkin et al., 2018). Recently, Geifman et al. (2020) have shown that when inputs are on the sphere, the Laplace kernel has the same decay as the NTK, which may suggest a similar conditioning of the optimization problem as for fully-connected networks, as discussed in Section 2.2. Denoting $\kappa_c(u) = e^{-c\sqrt{1-u}}$ so that $k_c(x, y) = \kappa_{c\sqrt{2}}(x^\top y)$, we may easily recover this result using Theorem 1 by noticing that $\kappa_c$ is infinitely differentiable around $-1$ and satisfies

$$\kappa_c(1 - t) = e^{-c\sqrt{t}} = 1 - c\sqrt{t} + O(t),$$

which yields the same decay $k^{-d}$ as the NTK. Geifman et al. (2020) also consider a heuristic generalization of the Laplace kernel with different exponents, $\kappa_{c,\gamma}(u) = e^{-c(1-u)^\gamma}$. Theorem 1 allows us to obtain a precise decay for this kernel as well using $\kappa_{c,\gamma}(1 - t) = 1 - ct^\gamma + O(t^{2\gamma})$, which is of the form $k^{-d-2\gamma+1}$ for non-integer $\gamma > 0$, and in particular approaches the limiting order of smoothness $(d - 1)/2$ when $\gamma \to 0$.[3]

**Deep kernels with step activations.** We saw in Section 3.2 that for ReLU activations, depth does not change the decay of the corresponding kernels. In contrast, when considering

---

[3]For $\kappa_c$ and $\kappa_{c,\gamma}$, the ability to differentiate expansions is straightforward since we have the exact expansion $\kappa_{c,\gamma}(u) = \sum_k c^k(1 - u)^{\gamma k}/k!$, which may be differentiated term-by-term.

step activations $\sigma(u) = \mathbb{1}\{u \geq 0\}$, we show in Appendix C.3 that approximation properties of the corresponding random neuron kernels (of the form $\kappa_0 \circ \cdots \circ \kappa_0$) improve with depth, leading to a decay $k^{-d-2\nu+1}$ with $\nu = 1/2^{L-1}$ for $L$ layers. This also leads to an RKHS which becomes as large as allowed (order of smoothness close to $(d-1)/2$) when $L \to \infty$. While this may suggest a benefit of depth, note that step activations make optimization hard for anything beyond a linear regime with random weights, since the gradients with respect to inner neurons vanish. Theorem 1 may also be applied to deep kernels with other positively homogeneous activations $\sigma_s(u) = \max(0, u)^s$ with $s \geq 2$, for which endpoint expansions easily follow from those of $\kappa_0$ or $\kappa_1$ through integration.

**Infinitely differentiable kernels.** Finally, we note that Theorem 1 shows that kernels associated to infinitely differentiable activations (which are themselves infinitely differentiable, see Daniely et al. (2016)[4]), as well as Gaussian kernels on the sphere of the form $e^{-c(1-x^\top y)}$, have faster decays than any polynomial. This results in a "small" RKHS that only contains smooth functions. See Azevedo & Menegatto (2014); Minh et al. (2006) for a more precise study of the decay for Gaussian kernels on the sphere.

## 4 Numerical experiments

We now present numerical experiments on synthetic and real data to illustrate our theory. Our code is available at `https://github.com/albietz/deep_shallow_kernel`.

**Synthetic experiments.** We consider randomly sampled inputs on the sphere $\mathbb{S}^3$ in 4 dimensions, and outputs generated according to the following target models, for an arbitrary $w \in \mathbb{S}^3$: $f_1^*(x) = \mathbb{1}\{w^\top x \geq 0.7\}$ and $f_2^*(x) = e^{-(1-w^\top x)^{3/2}} + e^{-(1+w^\top x)^{3/2}}$. Note that $f_1^*$ is discontinuous and thus not in the RKHS in general, while $f_2^*$ is in the RKHS of $\kappa_1$ (since it is even and has the same decay as $\kappa_1$ as discussed in Section 3.3). In Figure 1 we compare the quality of approximation for different kernels by examining generalization performance of ridge regression with exact kernels or random features. The regularization parameter $\lambda$ is optimized on $10\,000$ test datapoints on a logarithmic grid. In order to illustrate the difficulty of optimization due to a small optimal $\lambda$, which would also indicate slower convergence with gradient methods, we consider grids with $\lambda \geq \lambda_{\min}$, for two different choices of $\lambda_{\min}$. We see that all kernels provide a similar rate of approximation for a large enough grid, but when fixing a smaller optimization budget by taking a larger $\lambda_{\min}$, the NTK and Laplace kernels can achieve better performance for large sample size $n$, thanks to a slower eigenvalue decay of the covariance operator. Figure 1(right) shows that when using $m = \sqrt{n}$ random features (which can achieve optimal rates in some settings, see Rudi & Rosasco, 2017), the "shallow" ReLU network performs better than a three-layer version, despite having fewer weights. This suggests that in addition to providing no improvements to approximation in the infinite-width case, the kernel regimes for deep ReLU networks may even be worse than their two-layer counterparts in the finite-width setting.

**MNIST and Fashion-MNIST.** In Table 1, we consider the image classification datasets MNIST and Fashion-MNIST, which both consist of 60k training and 10k test images of size 28x28 with 10 output classes. We evaluate one-versus-all classifiers obtained by using kernel ridge regression by setting $y = 0.9$ for the correct label and $y = -0.1$ otherwise. We train on random subsets of 50k examples and use the remaining 10k examples for validation. We find that test accuracy is comparable for different numbers of layers in RF or NTK kernels, with a slightly poorer performance for the two-layer case likely due to parity constraints, in agreement with our theoretical result that the decay is the same for different $L$. There is a small decrease in accuracy for growing $L$, which may reflect changes in the decay constants or numerical errors when composing kernels. The slightly better performance of RF compared to NTK may suggest that these problems are relatively easy (*e.g.*, the regression function is smooth), so that a faster decay is preferable due to better adaptivity to smoothness.

---

[4]This requires the mild additional condition that each derivative of the activation is in $L^2$ w.r.t. the Gaussian measure.

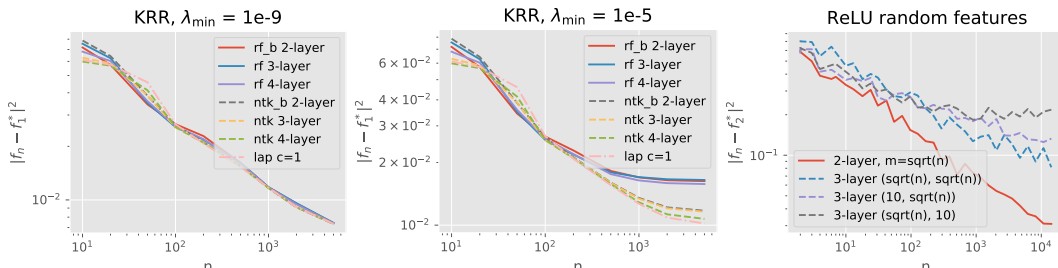

Figure 1: (left, middle) expected squared error vs sample size $n$ for kernel ridge regression estimators with different kernels on $f_1^*$ and with two different budgets on optimization difficulty $\lambda_{\min}$ (the minimum regularization parameter allowed). (right) ridge regression with one or two layers of random ReLU features on $f_2^*$, with different scalings of the number of "neurons" at each layer in terms of $n$.

Table 1: Test accuracies on MNIST (left) and Fashion-MNIST (right) for RF and NTK kernels with varying numbers of layers $L$. We use kernel ridge regression on 50k samples, with $\lambda$ optimized on a validation set of size 10k, and report mean and standard errors across 5 such random splits of the 60k training samples. For comparison, the Laplace kernel with $c = 1$ yields accuracies $98.39 \pm 0.02$ on MNIST and $90.38 \pm 0.06$ on F-MNIST.

<table>
<tr><td colspan="3" align="center">MNIST</td><td colspan="3" align="center">F-MNIST</td></tr>
<tr><td>L</td><td>RF</td><td>NTK</td><td>L</td><td>RF</td><td>NTK</td></tr>
<tr><td>2</td><td>$98.60 \pm 0.03$</td><td>$98.49 \pm 0.02$</td><td>2</td><td>$90.75 \pm 0.11$</td><td>$90.65 \pm 0.07$</td></tr>
<tr><td>3</td><td>$98.67 \pm 0.03$</td><td>$98.53 \pm 0.02$</td><td>3</td><td>$90.87 \pm 0.16$</td><td>$90.62 \pm 0.08$</td></tr>
<tr><td>4</td><td>$98.66 \pm 0.02$</td><td>$98.49 \pm 0.01$</td><td>4</td><td>$90.89 \pm 0.13$</td><td>$90.55 \pm 0.07$</td></tr>
<tr><td>5</td><td>$98.65 \pm 0.04$</td><td>$98.46 \pm 0.02$</td><td>5</td><td>$90.88 \pm 0.08$</td><td>$90.50 \pm 0.05$</td></tr>
</table>

## 5 Discussion

In this paper, we have analyzed the approximation properties of deep networks in kernel regimes, by studying eigenvalue decays of integral operators through differentiability properties of the kernel function. In particular, the decay is governed by the form of the function's (generalized) power series expansion around $\pm 1$, which remains the same for kernels arising from fully-connected ReLU networks of varying depths. This result suggests that the kernel approach is unsatisfactory for understanding the power of depth in fully-connected networks. In particular, it highlights the need to incorporate other regimes in the study of deep networks, such as the mean field regime (Chizat & Bach, 2018; Mei et al., 2018), and other settings with hierarchical structure (see, *e.g.*, Allen-Zhu & Li, 2020; Chen et al., 2020). We note that our results do not rule out benefits of depth for other network architectures in kernel regimes; for instance, depth may improve stability properties of convolutional kernels (Bietti & Mairal, 2019a;b), and a precise study of approximation for such kernels and its dependence on depth would also be of interest.

### Acknowledgments

The authors would like to thank David Holzmüller for finding an error in an earlier version of the paper, which led us to include the new assumption on differentiation of asymptotic expansions in Theorem 1. This work was funded in part by the French government under management of Agence Nationale de la Recherche as part of the "Investissements d'avenir" program, reference ANR-19-P3IA-0001 (PRAIRIE 3IA Institute). We also acknowledge support of the European Research Council (grant SEQUOIA 724063).

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

## A  BACKGROUND ON SPHERICAL HARMONICS

In this section, we provide some background on spherical harmonics needed for our study of approximation. See (Efthimiou & Frye, 2014; Atkinson & Han, 2012; Ismail, 2005) for references, as well as (Bach, 2017a, Appendix D). We consider inputs on the $d-1$ sphere $\mathbb{S}^{d-1} = \{x \in \mathbb{R}^d, \|x\| = 1\}$.

We recall some properties of the spherical harmonics $Y_{k,j}$ introduced in Section 2.2. For $j = 1, \ldots, N(d,k)$, where $N(d,k) = \frac{2k+d-2}{k}\binom{k+d-3}{d-2}$, the spherical harmonics $Y_{k,j}$ are homogeneous harmonic polynomials of degree $k$ that are orthonormal with respect to the uniform distribution $\tau$ on the $d$–1 sphere. The degree $k$ plays the role of an integer frequency, as in Fourier series, and the collection $\{Y_{k,j}, k \geq 0, j = 1, \ldots, N(d,k)\}$ forms an orthonormal basis of $L^2(\mathbb{S}^{d-1}, d\tau)$. As with Fourier series, there are tight connections between decay of coefficients in this basis w.r.t. $k$, and regularity/differentiability of functions, in this case differentiability on the sphere. This follows from the fact that spherical harmonics are eigenfunctions of the Laplace-Beltrami operator on the sphere $\Delta_{\mathbb{S}^{d-1}}$ (see Efthimiou & Frye, 2014, Proposition 4.5):

$$\Delta_{\mathbb{S}^{d-1}} Y_{k,j} = -k(k+d-2)Y_{k,j}. \tag{14}$$

For a given frequency $k$, we have the following addition formula:

$$\sum_{j=1}^{N(d,k)} Y_{k,j}(x)Y_{k,j}(y) = N(d,k)P_k(x^\top y), \tag{15}$$

where $P_k$ is the $k$-th Legendre polynomial in dimension $d$ (also known as Gegenbauer polynomial when using a different scaling), given by the Rodrigues formula:

$$P_k(t) = (-1/2)^k \frac{\Gamma(\frac{d-1}{2})}{\Gamma(k+\frac{d-1}{2})}(1-t^2)^{(3-d)/2}\left(\frac{d}{dt}\right)^k (1-t^2)^{k+(d-3)/2}. \tag{16}$$

Note that these may also be expressed using the hypergeometric function $_2F_1$ (see, *e.g.*, Ismail, 2005, Section 4.5), an expression we will use in proof of Theorem 1 (see the proof of Lemma 6).

The polynomials $P_k$ are orthogonal in $L^2([-1,1], d\nu)$ where the measure $d\nu$ is given by the weight function $d\nu(t) = (1-t^2)^{(d-3)/2}dt$, and we have

$$\int_{-1}^1 P_k^2(t)(1-t^2)^{(d-3)/2}dt = \frac{\omega_{d-1}}{\omega_{d-2}}\frac{1}{N(d,k)}, \tag{17}$$

where $\omega_{p-1} = \frac{2\pi^{p/2}}{\Gamma(p/2)}$ denotes the surface of the sphere $\mathbb{S}^{p-1}$ in $p$ dimensions. Using the addition formula (15) and orthogonality of spherical harmonics, we can show

$$\int P_j(w^\top x)P_k(w^\top y)d\tau(w) = \frac{\delta_{jk}}{N(d,k)}P_k(x^\top y) \tag{18}$$

We will use two other properties of Legendre polynomials, namely the following recurrence relation (Efthimiou & Frye, 2014, Eq. 4.36)

$$tP_k(t) = \frac{k}{2k+d-2}P_{k-1}(t) + \frac{k+d-2}{2k+d-2}P_{k+1}(t), \tag{19}$$

for $k \geq 1$, and for $k = 0$ we simply have $tP_0(t) = P_1(t)$, as well as the differential equation (see, *e.g.*, Efthimiou & Frye, 2014, Proposition 4.20):

$$(1 - t^2)P_k''(t) + (1 - d)tP_k'(t) + k(k + d - 2)P_k(t) = 0. \tag{20}$$

The Funk-Hecke formula is helpful for computing Fourier coefficients in the basis of spherical harmonics in terms of Legendre polynomials: for any $j = 1, \ldots, N(d, k)$, we have

$$\int f(x^\top y)Y_{k,j}(y)d\tau(y) = \frac{\omega_{d-2}}{\omega_{d-1}}Y_{k,j}(x)\int_{-1}^{1} f(t)P_k(t)(1 - t^2)^{(d-3)/2}dt. \tag{21}$$

For example, we may use this to obtain decompositions of dot-product kernels by computing Fourier coefficients of functions $\kappa(\langle x, \cdot \rangle)$. Indeed, denoting

$$\mu_k = \frac{\omega_{d-2}}{\omega_{d-1}}\int_{-1}^{1} \kappa(t)P_k(t)(1 - t^2)^{(d-3)/2}dt,$$

writing the decomposition of $\kappa(\langle x, \cdot \rangle)$ using (21) leads to the following Mercer decomposition of the kernel:

$$\kappa(x^\top y) = \sum_{k=0}^{\infty} \mu_k \sum_{j=1}^{N(d,k)} Y_{k,j}(x)Y_{k,j}(y) = \sum_{k=0}^{\infty} \mu_k N(d, k)P_k(x^\top y). \tag{22}$$

## B  PROOF OF THEOREM 1

The proof of Theorem 1, stated below in full as Theorem 7, proceeds as follows. We first derive an upper bound on the decay of $\kappa$ of the form $k^{-d-2\nu+3}$ (Lemma 5), which is weaker than the desired $k^{-d-2\nu+1}$, by exploiting regularity properties of $\kappa$ through integration by parts. The goal is then to apply this result on a function $\tilde{\kappa} = \kappa - \psi$, where $\psi$ is a function that allows us to "cancel" the leading terms in the expansions of $\kappa$, while being simple enough that it allows a precise estimate of its decay. In the proof of Theorem 7, we follow this strategy by considering $\psi$ as a sum of functions of the form $t \mapsto (1 - t^2)^\nu$ and $t \mapsto t(1 - t^2)^\nu$, for which we provide a precise computation of the decay in Lemma 6.

**Decay upper bound through regularity.** We begin by establishing a weak upper bound on the decay of $\kappa$ (Lemma 5) by leveraging its regularity up to the terms of order $(1 - t^2)^\nu$. This is achieved by iteratively applying the following integration by parts lemma, which is conceptually similar to integrating by parts on the sphere by leveraging the spherical Laplacian relation (14) in Appendix A, but directly uses properties of $\kappa$ and of Legendre polynomials instead (namely, the differential equation (20)). We note that the final statement in Theorem 1 on infinitely differentiable $\kappa$ directly follows from Lemma 5.

**Lemma 4** (Integration by parts lemma). *Let $\kappa : [-1, 1] \to \mathbb{R}$ be a function that is $C^\infty$ on $(-1, 1)$ and such that $\kappa'(t)(1 - t^2)^{1+\frac{d-3}{2}} = O(1)$. We have*

$$\int_{-1}^{1} \kappa(t)P_k(t)(1 - t^2)^{\frac{d-3}{2}}dt = \frac{1}{k(k + d - 2)}\left(-\kappa(t)(1 - t^2)^{1+\frac{d-3}{2}}P_k'(t)\Big|_{-1}^{1}\right. \tag{23}$$

$$\left. + \kappa'(t)(1 - t^2)^{1+\frac{d-3}{2}}P_k(t)\Big|_{-1}^{1} + \int_{-1}^{1} \tilde{\kappa}(t)P_k(t)(1 - t^2)^{(d-3)/2}dt\right), \tag{24}$$

*with $\tilde{\kappa}(t) = -\kappa''(t)(1 - t^2) + (d - 1)t\kappa'(t)$.*

*Proof.* In order to perform integration by parts, we use the following differential equation satisfied by Legendre polynomials (see, *e.g.*, Efthimiou & Frye, 2014, Proposition 4.20):

$$(1 - t^2)P_k''(t) + (1 - d)tP_k'(t) + k(k + d - 2)P_k(t) = 0. \tag{25}$$

Using this equation, we may write for $k \geq 1$,

$$\int_{-1}^{1} \kappa(t) P_k(t)(1-t^2)^{(d-3)/2} dt = \frac{1}{k(k+d-2)} \Big( (d-1) \int t\kappa(t) P_k'(t)(1-t^2)^{\frac{d-3}{2}} dt \tag{26}$$

$$- \int \kappa(t) P_k''(t)(1-t^2)^{1+\frac{d-3}{2}} dt \Big). \tag{27}$$

We may integrate the second term by parts using

$$\frac{d}{dt} \Big( \kappa(t)(1-t^2)^{1+\frac{d-3}{2}} \Big) = \kappa'(t)(1-t^2)^{1+\frac{d-3}{2}} - 2t(1+(d-3)/2)\kappa(t)(1-t^2)^{\frac{d-3}{2}}$$

$$= \kappa'(t)(1-t^2)^{1+\frac{d-3}{2}} - (d-1)t\kappa(t)(1-t^2)^{\frac{d-3}{2}}. \tag{28}$$

Noting that the first term in (26) cancels out with the integral resulting from the second term in (28), we then obtain

$$\int_{-1}^{1} \kappa(t) P_k(t)(1-t^2)^{(d-3)/2} dt = \frac{1}{k(k+d-2)} \Big( -\kappa(t)(1-t^2)^{1+\frac{d-3}{2}} P_k'(t) \Big|_{-1}^{1}$$

$$+ \int_{-1}^{1} \kappa'(t)(1-t^2)^{1+\frac{d-3}{2}} P_k'(t) dt \Big).$$

Integrating by parts once more, the second term becomes

$$\int_{-1}^{1} \kappa'(t)(1-t^2)^{1+\frac{d-3}{2}} P_k'(t) dt = \kappa'(t)(1-t^2)^{1+\frac{d-3}{2}} P_k(t) \Big|_{-1}^{1}$$

$$- \int_{-1}^{1} (\kappa''(t)(1-t^2) - (d-1)t\kappa'(t)) P_k(t)(1-t^2)^{(d-3)/2} dt. \tag{29}$$

The desired result follows. $\qquad\square$

**Lemma 5** (Weak upper bound on the decay). *Let $\kappa : [-1,1] \to \mathbb{R}$ be a function that is $C^\infty$ on $(-1,1)$ and has the following expansions around $\pm 1$ on its derivatives:*

$$\kappa^{(j)}(t) = p_{j,1}(1-t) + O((1-t)^{\nu-j}) \tag{30}$$

$$\kappa^{(j)}(t) = p_{j,-1}(1+t) + O((1+t)^{\nu-j}), \tag{31}$$

*for $t \in [-1,1]$ and $j \geq 0$, where $p_{j,1}, p_{j,-1}$ are polynomials and $\nu$ may be non-integer. Then the Legendre coefficients $\mu_k(\kappa)$ of $\kappa$ given in (8) satisfy*

$$\mu_k(\kappa) = O(k^{-d-2\nu+3}). \tag{32}$$

*Proof.* Let $f_0 := \kappa$ and for $j \geq 1$

$$f_j(t) := -f_{j-1}''(t)(1-t^2) + (d-1)f_{j-1}'(t). \tag{33}$$

Then $f_j$ is $C^\infty$ on $(-1,1)$ and has similar expansions to $\kappa$ of the form

$$f_j(t) = q_{j,1}(1-t) + O((1-t)^{\nu-j}) \tag{34}$$

$$f_j(t) = q_{j,-1}(1+t) + O((1+t)^{\nu-j}), \tag{35}$$

for some polynomials $q_{j,\pm 1}$. We may apply Lemma 4 repeatedly as long as the terms in brackets vanish, until we obtain, for $j = \lceil \nu + \frac{d-3}{2} \rceil - 1$,

$$\int_{-1}^{1} \kappa(t) P_k(t)(1-t^2)^{(d-3)/2} dt$$

$$= \frac{1}{(k(k+d-2))^{j+1}} \left( f_j'(t)(1-t^2)^{1+\frac{d-3}{2}} P_k(t) \Big|_{-1}^{1} + \int_{-1}^{1} f_{j+1}(t) P_k(t)(1-t^2)^{(d-3)/2} dt \right).$$

Given our choice for $j$, we have $f_j'(t)(1-t^2)^{1+\frac{d-3}{2}} = O(1)$, and $f_{j+1}(t)(1-t^2)^{(d-3)/2} = O((1-t^2)^{-1+\epsilon})$ for some $\epsilon > 0$. Since $P_k(t) \in [-1,1]$ for any $t \in [-1,1]$, we obtain $\mu_k(\kappa) = O(k^{-2(j+1)}) = O(k^{-d-2\nu+3})$. $\qquad\square$

**Precise decay for simple function.** We now provide precise decay estimates for functions of the form $t \mapsto (1 - t^2)^\nu$ and $t \mapsto t(1 - t^2)^\nu$, which will lead to the dominant terms in the decomposition of $\kappa$ in the main theorem.

**Lemma 6** (Decay for simple functions $\phi_\nu$ and $\bar{\phi}_\nu$). *Let $\phi_\nu(t) = (1 - t^2)^\nu$, with $\nu > 0$ non-integer, and let $\mu_k(\phi_\nu)$ denote its Legendre coefficients in d dimensions given by $\frac{\omega_{d-2}}{\omega_{d-1}} \int_{-1}^{1} (1 - t^2)^{\nu + (d-3)/2} P_k(t) dt$. We have*

- $\mu_k(\phi_\nu) = 0$ *if k is odd*
- $\mu_k(\phi_\nu) \sim C(d, \nu) k^{-d-2\nu-1}$ *for k even, $k \to \infty$, with $C(d, \nu)$ a constant.*

*Analogously, let $\bar{\phi}_\nu(t) := t(1 - t^2)^\nu$. We have*

- $\mu_k(\bar{\phi}_\nu) = 0$ *if k is even*
- $\mu_k(\bar{\phi}_\nu) \sim C(d, \nu) k^{-d-2\nu-1}$ *for k odd, $k \to \infty$, with $C(d, \nu)$ a constant.*

*Proof.* We recall the following representation of Legendre polynomials based on the hypergeometric function (*e.g.*, Ismail, 2005, Section 4.5):[5]

$$P_k(t) = {}_2F_1(-k, k + d - 2; (d - 1)/2; (1 - t)/2), \tag{36}$$

where the hypergeometric function is given in its generalized form by

$$_pF_q(a_1, \ldots, a_p; b_1, \ldots, b_q; x) = \sum_{s=0}^{\infty} \frac{(a_1)_s \cdots (a_p)_s}{(b_1)_s \cdots (b_q)_s} \frac{x^s}{s!}, \tag{37}$$

where $(a)_s = \Gamma(a + s)/\Gamma(a)$ is the rising factorial or Pochhammer symbol.

Using the above definitions and the integral representation of Beta functions, we then have

$$\int_{-1}^{1} (1 - t^2)^{\nu + \frac{d-3}{2}} P_k(t) dt = 2^{2\nu + d - 3} \int_{-1}^{1} \left(\frac{1 - t}{2}\right)^{\nu + \frac{d-3}{2}} \left(\frac{1 + t}{2}\right)^{\nu + \frac{d-3}{2}} P_k(t) dt$$

$$= 2^{2\nu + d - 3} \sum_{s=0}^{k} \frac{(-k)_s (d - 2 + k)_s}{\left(\frac{d-1}{2}\right)_s s!} \int_{-1}^{1} \left(\frac{1 - t}{2}\right)^{\nu + \frac{d-3}{2} + s} \left(\frac{1 + t}{2}\right)^{\nu + \frac{d-3}{2}} dt$$

$$= 2^{2\nu + d - 2} \sum_{s=0}^{k} \frac{(-k)_s (d - 2 + k)_s}{\left(\frac{d-1}{2}\right)_s s!} \int_{0}^{1} (1 - x)^{\nu + \frac{d-3}{2} + s} x^{\nu + \frac{d-3}{2}} dx$$

$$= 2^{2\nu + d - 2} \sum_{s=0}^{k} \frac{(-k)_s (d - 2 + k)_s}{\left(\frac{d-1}{2}\right)_s s!} \frac{\Gamma(\nu + s + \frac{d-1}{2})\Gamma(\nu + \frac{d-1}{2})}{\Gamma(2\nu + s + d - 1)}$$

$$= 2^{2\nu + d - 2} \frac{\Gamma(\nu + \frac{d-1}{2})^2}{\Gamma(2\nu + d - 1)} \sum_{s=0}^{k} \frac{(-k)_s (d - 2 + k)_s (\nu + \frac{d-1}{2})_s}{\left(\frac{d-1}{2}\right)_s (2\nu + d - 1)_s s!}$$

$$= 2^{2\nu + d - 2} \frac{\Gamma(\nu + \frac{d-1}{2})^2}{\Gamma(2\nu + d - 1)} {}_3F_2 \left(\begin{matrix} -k, k + d - 2, \nu + (d - 1)/2 \\ (d - 1)/2, 2\nu + d - 1 \end{matrix} \middle| 1\right).$$

Now, we use Watson's theorem (*e.g.*, Ismail, 2005, Eq. (1.4.12)), which states that

$$_3F_2 \left(\begin{matrix} a, b, c \\ (a + b + 1)/2, 2c \end{matrix} \middle| 1\right) = \frac{\Gamma(\frac{1}{2})\Gamma(c + \frac{1}{2})\Gamma(\frac{a+b+1}{2})\Gamma(c + \frac{1-a-b}{2})}{\Gamma(\frac{a+1}{2})\Gamma(\frac{b+1}{2})\Gamma(c + \frac{1-a}{2})}. \tag{38}$$

We remark that with $a = -k, b = k + d - 2, c = \nu + (d - 1)/2$, our expression above is of the form of Watson's theorem, and we may thus evaluate $\mu_k$ in closed form. Indeed, we have

$$_3F_2 \left(\begin{matrix} -k, k + d - 2, \nu + (d - 1)/2 \\ (d - 1)/2, 2\nu + d - 1 \end{matrix} \middle| 1\right) = \frac{\Gamma(\frac{1}{2})\Gamma(\nu + \frac{d}{2})\Gamma(\frac{d-1}{2})\Gamma(\nu + 1)}{\Gamma(\frac{1-k}{2})\Gamma(\frac{d+k-1}{2})\Gamma(\nu + \frac{k}{2} + \frac{d}{2})\Gamma(\nu + 1 - \frac{k}{2})}. \tag{39}$$

---

[5]Here we normalize such that $P_k(1) = 1$ as is standard for Legendre polynomials, in contrast to (Ismail, 2005) where the standard Jacobi/Gegenbauer normalization is used.

When $k$ is odd, then $(1 - k)/2$ is a non-positive integer so that the denominator is infinite and thus $\mu_k$ vanishes. We assume from now on that $k$ is even, making the denominator is finite. Using the following relation, for $\epsilon \notin \mathbb{Z}$ and an integer $n$:

$$\frac{\Gamma(1 + \epsilon)}{\Gamma(\epsilon - n)} = \epsilon(\epsilon - 1) \cdots (\epsilon - n) = (-1)^{n-1} \frac{\Gamma(n + 1 - \epsilon)}{\Gamma(-\epsilon)}, \tag{40}$$

we may then rewrite

$$_3F_2\left(\begin{matrix} -k, k + d - 2, \nu + (d-1)/2 \\ (d-1)/2, 2\nu + d - 1 \end{matrix}\bigg|1\right) = \frac{\Gamma(\nu + \frac{d}{2})\Gamma(\frac{d-1}{2})\Gamma(\nu + 1)}{\Gamma(-\frac{1}{2})\Gamma(\nu + 2)\Gamma(-\nu - 1)} \frac{\Gamma(\frac{k+1}{2})\Gamma(\frac{k}{2} - \nu)}{\Gamma(\frac{d+k-1}{2})\Gamma(\nu + \frac{k}{2} + \frac{d}{2})}. \tag{41}$$

When $k \to \infty$, Stirling's formula $\Gamma(x) \sim x^{x - \frac{1}{2}} e^{-x} \sqrt{2\pi}$ yields the equivalent

$$\frac{\Gamma(\frac{k+1}{2})\Gamma(\frac{k}{2} - \nu)}{\Gamma(\frac{d+k-1}{2})\Gamma(\nu + \frac{k}{2} + \frac{d}{2})} \sim \left(\frac{k}{2}\right)^{-d - 2\nu + 1}. \tag{42}$$

This yields

$$\mu_k \sim C(d, \nu) k^{-d - 2\nu + 1}, \tag{43}$$

with

$$C(d, \nu) = 2^{2\nu + d - 2} \frac{\omega_{d-2}}{\omega_{d-1}} \frac{\Gamma(\nu + \frac{d-1}{2})^2}{\Gamma(2\nu + d - 1)} \frac{\Gamma(\nu + \frac{d}{2})\Gamma(\frac{d-1}{2})\Gamma(\nu + 1)}{\Gamma(-\frac{1}{2})\Gamma(\nu + 2)\Gamma(-\nu - 1)} (1/2)^{-d - 2\nu + 1}. \tag{44}$$

**Decay for $\bar{\phi}_\nu$.** The decay for $\bar{\phi}_\nu$ follows from the decay of $\phi_\nu$ and the recurrence relation (Efthimiou & Frye, 2014, Eq. (4.36))

$$t P_k(t) = \frac{k}{2k + d - 2} P_{k-1}(t) + \frac{k + d - 2}{2k + d - 2} P_{k+1}(t), \tag{45}$$

which ensures the same decay with a change parity. $\qquad\square$

**Final theorem.** We are now ready to prove our main theorem, which differs from the simplified statement of Theorem 1 by the technical assumption that only a finite number $r$ of terms of order between $\nu$ and $\nu + 1$ are present in the series expansions around $\pm 1$.

**Theorem 7** (Main theorem, full version). *Let $\kappa : [-1, 1] \to \mathbb{R}$ be a function that is $C^\infty$ on $(-1, 1)$ and has the following expansions around $\pm 1$:*

$$\kappa(t) = p_1(1 - t) + \sum_{j=1}^{r} c_{j,1}(1 - t)^{\nu_j} + O((1 - t)^{\nu_1 + 1 + \epsilon}) \tag{46}$$

$$\kappa(t) = p_{-1}(1 + t) + \sum_{j=1}^{r} c_{j,-1}(1 + t)^{\nu_j} + O((1 + t)^{\nu_1 + 1 + \epsilon}), \tag{47}$$

*for $t \in [-1, 1]$, where $p_1, p_{-1}$ are polynomials and $0 < \nu_1 < \ldots < \nu_r$ are not integers and $0 < \epsilon < \nu_2 - \nu_1$. We also assume that the derivatives $\kappa^{(s)}$ of $\kappa$ have the following expansions:*

$$\kappa^{(s)}(t) = p_{s,1}(1 - t) + (-1)^s \sum_{j=1}^{r} c_{j,1} \frac{\Gamma(\nu_j + 1)}{\Gamma(\nu_j + 1 - s)} (1 - t)^{\nu_j - s} + O((1 - t)^{\nu_1 + 1 + \epsilon - s}) \tag{48}$$

$$\kappa^{(s)}(t) = p_{s,-1}(1 + t) + \sum_{j=1}^{r} c_{j,-1} \frac{\Gamma(\nu_j + 1)}{\Gamma(\nu_j + 1 - s)} (1 + t)^{\nu_j - s} + O((1 + t)^{\nu_1 + 1 + \epsilon - s}), \tag{49}$$

*for some polynomials $p_{s,\pm 1}$. Then we have, for an absolute constant $C(d, \nu_1)$ depending only on $d$ and $\nu_1$,*

- *For $k$ even, if $c_{\nu_1, 1} \neq -c_{1,-1}$: $\mu_k \sim (c_{1,1} + c_{1,-1}) C(d, \nu_1) k^{-d - 2\nu_1 + 1}$;*
- *For $k$ even, if $c_{1,1} = -c_{1,-1}$: $\mu_k = o(k^{-d - 2\nu_1 + 1})$;*

- *For $k$ odd, if $c_{1,1} \neq c_{1,-1}$: $\mu_k \sim (c_{1,1} - c_{1,-1})C(d, \nu_1)k^{-d-2\nu_1+1}$.*
- *For $k$ odd, if $c_{1,1} = c_{1,-1}$: $\mu_k = o(k^{-d-2\nu_1+1})$.*

*Proof.* Define the functions

$$\psi_j(t) = c_{j,1} \frac{\phi_{\nu_j}(t) + \bar{\phi}_{\nu_j}(t)}{2^{\nu_j+1}} + c_{j,-1} \frac{\phi_{\nu_j}(t) - \bar{\phi}_{\nu_j}(t)}{2^{\nu_j+1}} \tag{50}$$

$$= \frac{c_{j,1} + c_{j,-1}}{2^{\nu_j+1}} \phi_{\nu_j}(t) + \frac{c_{j,1} - c_{j,-1}}{2^{\nu_j+1}} \bar{\phi}_{\nu_j}(t), \tag{51}$$

for $j = 1, \ldots, r$, where $\phi_\nu, \bar{\phi}_\nu$ are defined in Lemma 6. We have the asymptotic expansions:[6]

$$\psi_1(t) = c_{1,1}(1-t)^{\nu_1} - \frac{(1+\nu_1)c_{1,1} + c_{1,-1}}{2}(1-t)^{\nu_1+1} + O((1-t)^{\nu_1+1+\epsilon})$$

$$\psi_1(t) = c_{1,-1}(1+t)^{\nu_1} + \frac{c_{1,1} - (1+\nu)c_{1,-1}}{2}(1+t)^{\nu_1+1} + O((1+t)^{\nu_1+1+\epsilon}),$$

and for $j \geq 2$,

$$\psi_j(t) = c_{j,1}(1-t)^{\nu_j} + O((1-t)^{\nu_1+1+\epsilon})$$

$$\psi_j(t) = c_{j,-1}(1+t)^{\nu_j} + O((1+t)^{\nu_1+1+\epsilon}).$$

Define additionally $\psi_{r+1}$ the same way as the other $\psi_j$, with $\nu_{r+1} = \nu_1 + 1$, $c_{r+1,1} = ((1+\nu_1)c_{1,1} + c_{1,-1})/2$, and $c_{r+1,-1} = -(c_{1,1} - (1+\nu)c_{1,-1})/2$, which satisfies a similar asymptotic expansion as the above ones for $j \geq 2$. One can check that the derivatives of the $\psi_j$ can be expanded with the derivatives of the expansions above. Then, defining $\tilde{\kappa} = \kappa - \sum_{j=1}^{r+1} \psi_j$, we have for $s \geq 0$,

$$\tilde{\kappa}^{(s)}(t) = p_{s,1}(1-t) + O((1-t)^{\nu_1+1+\epsilon-s}) \tag{52}$$

$$\tilde{\kappa}^{(s)}(t) = p_{s,-1}(1+t) + O((1+t)^{\nu_1+1+\epsilon-s}). \tag{53}$$

The functions $\psi_j$ satisfy

$$\mu_k(\psi_j) = \begin{cases} \frac{c_{j,1}+c_{j,-1}}{2^{\nu_j+1}} \mu_k(\phi_{\nu_j}), & \text{if } k \text{ even,} \\ \frac{c_{j,1}-c_{j,-1}}{2^{\nu_j+1}} \mu_k(\bar{\phi}_{\nu_j}), & \text{if } k \text{ odd.} \end{cases} \tag{54}$$

By Lemma 5, we have

$$\mu_k(\kappa) = \mu_k(\tilde{\kappa}) + \sum_{j=1}^{r} \mu_k(\psi_j) \tag{55}$$

$$= \sum_{j=1}^{r} \mu_k(\psi_j) + O(k^{-d-2(\nu_1+1+\epsilon)+3}) \tag{56}$$

$$= \sum_{j=1}^{r} \mu_k(\psi_j) + o(k^{-d-2\nu_1+1}). \tag{57}$$

The result then follows from Lemma 6, with a constant $C(d, \nu_1)/2^{\nu_1+1}$, where $C(d, \nu_1)$ is given by the proof of Lemma 6. $\square$

### B.1 DIMENSION-FREE DESCRIPTION

While our above description of the RKHS depends on the dimension $d$, in some cases a dimension-free description given by Taylor coefficients of the kernel $\kappa$ at 0 may be useful, for instance for the study of kernel methods in certain high-dimensional regimes (*e.g.*, El Karoui, 2010; Ghorbani et al., 2019; Liang et al., 2020). Here we remark that such coefficients and

---

[6]These are obtained by writing $\psi_j(t) = (a + bt)(1 + t)^\nu(1 - t)^\nu$ and computing, *e.g.*, the first two terms in the analytic expansion of $t \mapsto (a + bt)(1 + t)^\nu$ around 1.

their decay may be recovered from the Legendre coefficients in $d$ dimensions, by taking high-dimensional limits $d \to \infty$. We illustrate this on the functions $\phi_\nu(t) = (1 - t^2)^\nu$, for which Lemma 6 provides precise estimates of the Legendre coefficients $\mu_{k,d}(\phi_\nu)$ in $d$ dimensions (this only serves as an instructive illustration, since in this case Taylor coefficients may be computed directly through a power series expansion of $\phi_\nu$ using the Binomial formula).

**Lemma 8** (Recovering Taylor coefficients of $\phi_\nu$ through high-dimensional limits)**.**
Let $b_k(\phi_\nu) := \frac{\phi_\nu^{(k)}}{k!}$ for some non-integer $\nu > 0$. For $k$ even, we have

$$b_k(\phi_\nu) = C_\nu 2^k \frac{\Gamma(\frac{k+1}{2})\Gamma(\frac{k}{2} - \nu)}{\Gamma(k+1)},$$

for a constant $C_\nu$ depending only on $\nu$. This leads to an equivalent $b_k \sim C'_\nu k^{-\nu-1}$ for $k \to \infty$ with $k$ even.

*Proof.* Assume throughout that $k$ is even. Recall the expression of the Legendre coefficients $\mu_{k,d}(\phi_\nu)$ of $\phi_\nu$ in $d$ dimensions (we include $d$ as a subscript for more clarity here) from the proof of Lemma 6:

$$\mu_{k,d}(\phi_\nu) = \frac{\omega_{d-2}}{\omega_{d-1}} \int_{-1}^1 \kappa(t) P_{k,d}(t)(1-t^2)^{\frac{d-3}{2}} dt \tag{58}$$

$$= 2^{2\nu+d-2} \frac{\omega_{d-2}}{\omega_{d-1}} \frac{\Gamma(\nu + \frac{d-1}{2})^2}{\Gamma(2\nu+d-1)} \frac{\Gamma(\nu + \frac{d}{2})\Gamma(\frac{d-1}{2})\Gamma(\nu+1)}{\Gamma(-\frac{1}{2})\Gamma(\nu+2)\Gamma(-\nu-1)} \frac{\Gamma(\frac{k+1}{2})\Gamma(\frac{k}{2} - \nu)}{\Gamma(\frac{d+k-1}{2})\Gamma(\nu + \frac{k}{2} + \frac{d}{2})}. \tag{59}$$

Now, note that when $d$ is large enough compared to $k$, we may use the Rodrigues formula (16) and integration by parts to obtain the following alternative expression:

$$\mu_{k,d}(\phi_\nu) = 2^{-k} \frac{\omega_{d-2}}{\omega_{d-1}} \frac{\Gamma(\frac{d-1}{2})}{\Gamma(k + \frac{d-1}{2})} \int_{-1}^1 \phi_\nu^{(k)}(t)(1-t^2)^{k+\frac{d-3}{2}} dt$$

Following similar arguments to Ghorbani et al. (2019), we may then use dominated convergence to show:

$$\frac{\Gamma(\frac{d}{2})}{\sqrt{\pi}\Gamma(\frac{d-1}{2})} \int_{-1}^1 \phi_\nu^{(k)}(t)(1-t^2)^{k+\frac{d-3}{2}} dt \to \phi_\nu^{(k)}(0) \quad \text{as } d \to \infty.$$

Indeed, $\frac{\Gamma(\frac{d}{2})}{\sqrt{\pi}\Gamma(\frac{d-1}{2})}(1-t^2)^{(d-3)/2}$ is a probability density that approaches a Dirac mass at 0 when $d \to \infty$. This yields

$$b_k(\phi_\nu) = \frac{\phi_\nu^{(k)}}{k!} = \lim_{d\to\infty} 2^k \frac{\omega_{d-1}}{\omega_{d-2}} \frac{\Gamma(\frac{d}{2})\Gamma(k + \frac{d-1}{2})}{\sqrt{\pi}\Gamma(\frac{d-1}{2})\Gamma(\frac{d-1}{2})\Gamma(k+1)} \mu_{k,d}(\phi_\nu).$$

Plugging (59) and using Stirling's formula to take limits $d \to \infty$ yields

$$b_k(\phi_\nu) = C_\nu 2^k \frac{\Gamma(\frac{k+1}{2})\Gamma(\frac{k}{2} - \nu)}{\Gamma(k+1)},$$

where $C_\nu$ only depends on $\nu$. Using Stirling's formula once again yields the desired equivalent $b_k(\phi_\nu) \sim C'_\nu k^{-\nu-1}$ for $k \to \infty$, $k$ even, with a different constant $C'_\nu$. $\qquad\square$

We note that a similar asymptotic equivalent holds for $b_k(\bar{\phi}_\nu)$ for $k$ odd. The next result leverages this to derive asymptotic decays of $b_k(\kappa)$ for any $\kappa$ of the form $\kappa(u) = \sum_{k\geq 0} b_k(\kappa) u^k$ satisfying similar conditions as in Theorem 7.

**Corollary 9** (Taylor coefficients of $\kappa$)**.** *Let $\kappa : [-1, 1] \to \mathbb{R}$ be a function admitting a power series expansion $\kappa(u) = \sum_{k\geq 0} b_k u^k$, with the following expansions around $\pm 1$:*

$$\kappa(t) = p_1(1-t) + \sum_{j=1}^r c_{j,1}(1-t)^{\nu_j} + O((1-t)^{\lceil \nu_1 \rceil + 1}) \tag{60}$$

$$\kappa(t) = p_{-1}(1+t) + \sum_{j=1}^r c_{j,-1}(1+t)^{\nu_j} + O((1+t)^{\lceil \nu_1 \rceil + 1}), \tag{61}$$

*for* $t \in [-1, 1]$, *where* $p_1, p_{-1}$ *are polynomials and* $0 < \nu_1 < \ldots < \nu_r$ *are not integers and* $0 < \epsilon < \nu_2 - \nu_1$. *Then we have, for an absolute constant* $C(\nu_1)$ *depending only on* $\nu_1$,

- *For $k$ even, if $c_{\nu_1,1} \neq -c_{\nu_1,-1}$: $b_k \sim (c_{\nu_1,1} + c_{\nu_1,-1})C(\nu_1)k^{-\nu_1-1}$;*
- *For $k$ even, if $c_{\nu_1,1} = -c_{\nu_1,-1}$: $b_k = o(k^{-\nu_1-1})$;*
- *For $k$ odd, if $c_{\nu_1,1} \neq c_{\nu_1,-1}$: $b_k \sim (c_{\nu_1,1} - c_{\nu_1,-1})C(\nu_1)k^{-\nu_1-1}$.*
- *For $k$ odd, if $c_{\nu_1,1} = c_{\nu_1,-1}$: $b_k = o(k^{-\nu_1-1})$.*

*Proof.* As in the proof of Theorem 7, we may construct a function $\psi = \sum_j \alpha_j \phi_{\nu_j} + \bar{\alpha}_j \bar{\phi}_{\nu_j}$, with $\alpha_1 = \frac{c_{1,1}+c_{1,-1}}{2^{\nu_1}+1}$, $\bar{\alpha}_1 = \frac{c_{1,1}-c_{1,-1}}{2^{\nu_1}+1}$ for $j = 1$, the other terms being of higher orders $\nu_j > \nu_1$, such that $\tilde{\kappa} := \kappa - \psi$ (which is also a power series with convergence radius $\geq 1$) satisfies

$$\tilde{\kappa}(t) = p_1(1-t) + O((1-t)^{\lceil \nu_1 \rceil + 1}) \tag{62}$$

$$\tilde{\kappa}(t) = p_{-1}(1+t) + O((1+t)^{\lceil \nu_1 \rceil + 1}), \tag{63}$$

It follows that $\tilde{\kappa}^{(\lceil \nu_1 \rceil + 1)}(1)$ is bounded, so that the Taylor coefficients of $\tilde{\kappa}$, denoted $b_k(\tilde{\kappa})$, satisfy

$$b_k(\tilde{\kappa}) = o(k^{-\lceil \nu_1 \rceil - 1}) = o(k^{-\nu_1 - 1}).$$

The result then follows from Lemma 8 by using the decays of $b_k(\phi_\nu)$ and $b_k(\bar{\phi}_\nu)$.

$\square$

## C   OTHER PROOFS

In this section, we provide the proofs for results in Section 3.3 related to obtaining power series expansions (with generalized exponents) of kernels arising from deep networks, which leads to the corresponding decays by Theorem 1. We note that for the kernels considered in this section, we can differentiate the expansions since the kernel function is $\Delta$-analytic (see Chen & Xu, 2021, Theorem 7), so that the technical assumption in Theorem 1 is verified.

### C.1   PROOF OF COROLLARY 2

*Proof.* Let $\kappa^\ell := \underbrace{\kappa_1 \circ \cdots \circ \kappa_1}_{\ell-1 \text{ times}} = \kappa_{\text{RF}}^\ell$. We have

$$\kappa_1(1-t) = 1 - t + ct^{3/2} + o(t^{3/2}), \quad c := \frac{2\sqrt{2}}{3\pi}.$$

We now show by induction that $\kappa^\ell(1-t) = 1 - t + a_\ell t^{3/2} + o(t^{3/2})$, with $a_\ell = (\ell-1)c$. This is obviously true for $\ell = 2$ since $\kappa^\ell = \kappa_1$, and for $\ell \geq 3$ we have

$$\begin{aligned}
\kappa^\ell(1-t) &= \kappa_1(\kappa^{\ell-1}(1-t)) \\
&= \kappa_1(1 - t + a_{\ell-1}t^{3/2} + o(t^{3/2})) \\
&= 1 - (t - a_{\ell-1}t^{3/2} + o(t^{3/2})) + c(t + O(t^{3/2}))^{3/2} + o(O(t)^{3/2}) \\
&= 1 - t + a_{\ell-1}t^{3/2} + ct^{3/2}(1 + O(t^{1/2}))^{3/2} + o(t^{3/2}) \\
&= 1 - t + a_{\ell-1}t^{3/2} + ct^{3/2}(1 + O(t^{1/2})) + o(t^{3/2}) \\
&= 1 - t + a_\ell t^{3/2} + o(t^{3/2}),
\end{aligned}$$

which proves the result.

Around $-1$, we know that

$$\kappa_1(-1+t) = ct^{3/2} + o(t^{3/2}).$$

We then have $\kappa^\ell(-1+t) = b_\ell + c_\ell t^{3/2} + o(t^{3/2})$, with $0 \leq b_\ell < 1$ and $0 < c_\ell \leq c$ (and the upper bound is strict for $\ell \geq 3$). Indeed, this is true for $\ell = 2$, and for $\ell \geq 3$ we have,

for $t > 0$,

$$
\begin{aligned}
\kappa^\ell(-1 + t) &= \kappa_1(\kappa^{\ell-1}(-1 + t)) \\
&= \kappa_1(b_{\ell-1} + c_{\ell-1}t^{3/2} + o(t^{3/2})) \\
&= \kappa_1(b_{\ell-1}) + \kappa_1'(b_{\ell-1})c_{\ell-1}t^{3/2} + o(t^{3/2}).
\end{aligned}
$$

Now, note that $\kappa_1$ and $\kappa_1'$ are both positive and strictly increasing on $[0, 1]$, with $\kappa_1(1) = \kappa_1'(1) = 1$. Thus, we have $b_\ell = \kappa_1(b_{\ell-1}) \in (0, 1)$, and $c_\ell = \kappa_1'(a_{\ell-1})c_{\ell-1} < c_{\ell-1}$, thus completing the proof.

Since $c_\ell$ is bounded while $a_\ell$ grows linearly with $\ell$, the constants in front of the asymptotic decay $k^{-d-2}$ grow linearly with $\ell$.

$\qquad\qquad\qquad\qquad\qquad\qquad\qquad\qquad\qquad\qquad\qquad\qquad\qquad\qquad\qquad\qquad\quad\square$

### C.2    Proof of Corollary 3

*Proof.* We show by induction that $\kappa_{\mathrm{NTK}}^\ell$ as defined in (7) satisfies

$$
\kappa_{\mathrm{NTK}}^\ell(1 - t) = \ell - \left(\sum_{s=1}^{\ell-1} s\right)ct^{1/2} + o(t^{1/2}), \qquad c := \frac{\sqrt{2}}{\pi}.
$$

For $\ell = 2$ we have $\kappa_{\mathrm{NTK}}^\ell(u) = u\kappa_0(u) + \kappa_1(u)$, so that

$$
\kappa_{\mathrm{NTK}}^2(1 - t) = (1 - t)(1 - ct^{1/2} + o(t^{1/2})) + 1 + O(t) = 2 - ct^{1/2} + o(t^{1/2}).
$$

By induction, for $\ell \geq 3$, we have $\kappa_{\mathrm{NTK}}^\ell(u) = \kappa_{\mathrm{NTK}}^{\ell-1}(u)\kappa_0(\kappa^{\ell-1}(u)) + \kappa^\ell(u)$, with $\kappa^\ell$ as in the proof of Corollary 2, which hence satisfies $\kappa^\ell(1 - t) = 1 - t + o(t)$ for all $\ell \geq 2$. We then have

$$
\begin{aligned}
\kappa_0(\kappa^{\ell-1}(1 - t)) &= \kappa_0(1 - t + o(t)) \\
&= 1 - c(t + o(t))^{1/2} + o(t^{1/2}) \\
&= 1 - ct^{1/2}(1 + o(t^{1/2})) + o(t^{1/2}) \\
&= 1 - ct^{1/2} + o(t^{1/2}).
\end{aligned}
$$

This yields

$$
\begin{aligned}
\kappa_{\mathrm{NTK}}^\ell(1 - t) &= (\ell - 1 - (\sum_{s=1}^{\ell-2} s)ct^{1/2} + o(t^{1/2}))(1 - ct^{1/2} + o(t^{1/2})) + 1 + O(t) \\
&= \ell - (\sum_{s=1}^{\ell-1} s)ct^{1/2} + o(t^{1/2}) \\
&= \ell - \frac{\ell(\ell - 1)}{2}ct^{1/2} + o(t^{1/2}),
\end{aligned}
$$

which proves the claim for the expansion around $+1$.

Around -1, recall the expansion from the proof of Corollary 2, $\kappa^\ell(-1 + t) = b_\ell + O(t^{3/2})$, with $0 \leq b_\ell < 1$. For $\ell = 2$, we have

$$
\kappa_{\mathrm{NTK}}^2(-1 + t) = (-1 + t)(ct^{1/2} + o(t^{1/2})) + b_2 + o(t^{1/2}) = b_2 - ct^{1/2} + o(t^{1/2}).
$$

Note also that for $\ell \geq 2$,

$$
\kappa_0(\kappa^\ell(-1 + t)) = \kappa_0(b_\ell + O(t^{3/2})) = \kappa_0(b_\ell) + O(t^{3/2}),
$$

since $\kappa_0'(b_\ell)$ is finite for $b_\ell < 1$. We also have $\kappa_0(b_\ell) \in (0, 1)$ since $\kappa_0$ is positive and strictly increasing on $[0, 1]$ with $\kappa_0(1) = 1$. Then, by an easy induction, we obtain

$$
\kappa_{\mathrm{NTK}}^\ell(-1 + t) = a_\ell - c_\ell t^{1/2} + o(t^{1/2}),
$$

with $a_\ell \leq \ell$ and $0 < c_\ell < c$.

Similar to the case of the RF kernel, the constant in front of $t^{1/2}$ grows with $\ell^2$ for the expansion around $+1$ but is bounded for the expansion around $-1$, so that the final constants in front of the asymptotic decay $k^{-d}$ grow quadratically with $\ell$. However, they grow linearly with $\ell$ when considering the NTK normalized by $\ell$, $\tilde\kappa^\ell = \kappa_{\mathrm{NTK}}^\ell/\ell$, which then satisfies $\tilde\kappa^\ell(1) = 1$.

$\square$

## C.3 Deep networks with step activations

In this section, we study the decay of the random weight kernel arising from deep networks with step activations, as presented in Section 3.3. For an $L$-layer network, this kernel is of the form $\kappa_s^L := \underbrace{\kappa_0 \circ \cdots \circ \kappa_0}_{L-1 \text{ times}}$.

**Corollary 10.** *$\kappa_s^L$ has a decay $k^{-d-2\nu_L+1}$ with $\nu_L = 1/2^{L-1}$ for $L$ layers.*

*Proof.* We show by induction that we have, for $\ell \geq 2$,

$$\kappa_s^\ell(1-t) = 1 - c^{\sum_{j=0}^{\ell-1} 2^{-j}} t^{1/2^{\ell-1}} + o(t^{1/2^{\ell-1}}),$$

with $c := \frac{\sqrt{2}}{\pi}$. This is true for $\ell = 2$ due to the expansion for $\kappa_0$. Now assume it holds for $\ell \geq 2$. We have

$$\begin{aligned}
\kappa_s^{\ell+1}(1-t) &= \kappa_0(\kappa_s^\ell(1-t)) \\
&= \kappa_0\left(1 - c^{\sum_{j=0}^{\ell-1} 2^{-j}} t^{1/2^{\ell-1}} + o(t^{1/2^{\ell-1}})\right) \\
&= 1 - c\left(c^{\sum_{j=0}^{\ell-1} 2^{-j}} t^{1/2^{\ell-1}} + o(t^{1/2^{\ell-1}})\right)^{1/2} + o(o(t^{1/2^{\ell-1}})^{1/2}) \\
&= 1 - c^{\sum_{j=0}^{\ell} 2^{-j}} t^{1/2^\ell}(1 + o(1)) + o(t^{1/2^\ell}) \\
&= 1 - c^{\sum_{j=0}^{\ell} 2^{-j}} t^{1/2^\ell} + o(t^{1/2^\ell}),
\end{aligned}$$

proving the desired claim.

Around $-1$, we have $\kappa_0(-1+t) = ct^{1/2} + o(t^{1/2})$, and for $\ell \geq 3$, $\kappa_s^\ell(-1+t) = a_\ell + O(t^{1/2})$, by an easy induction using the fact that $\kappa_0([0,1)) \subset (0,1)$ and $\kappa_0$ is smooth on $[0,1)$. Thus the behavior around $-1$ does not affect the decay of $\kappa_s^\ell$ for $\ell \geq 3$, and Theorem 1 leads to the desired decay, with a constant that only depends on $\ell$ through $c^{\sum_{j=0}^{\ell-1} 2^{-j}}$, which lies in the interval $[c^2, c]$ for any $\ell$. $\square$

