# OpenReview forum: "Deep Equals Shallow for ReLU Networks in Kernel Regimes"
_ICLR.cc/2021/Conference — ICLR 2021 Poster_

### Official Review · AnonReviewer1 · 2020-10-20
**A nice result in the field of NTK and deep learning**

**Rating:** 9
**Confidence:** 5

**Review:**


Recently, there are a large number of deep learning theory papers related to the property of neural tangent kernel. This paper shows that for ReLU, the kernels derived from deep fully-connected networks have the same approx. properties as their shallow two-layer counterpart. This highlights the limitation of the kernel framework for understanding the benefits of deep networks from such perspective.

I really like the idea of this paper. This paper is very well-written. I verified the proofs, it looks correct. I think this paper should be clearly accepted.

Here are some minor comments about references.

In page 1, several references should be added into NTK literature

Sanjeev Arora, Simon Du, Wei Hu, Zhiyuan Li, and Ruosong Wang.
Fine-grained analysis of optimization and generalization for overparameterized two-layer neural networks (ICML)

Zeyuan Allen-Zhu, Yuanzhi Li, and Yingyu Liang. Learning and generalization in overparameterized neural networks, going beyond two layers. (NeurIPS)

Zeyuan Allen-Zhu, Yuanzhi Li, and Zhao Song. On the convergence rate of training recurrent neural networks. (NeurIPS)

Jan van den Brand, Binghui Peng, Zhao Song, Omri Weinstein. Training (Overparametrized) Neural Networks in Near-Linear Time. (ITCS)

Zhao Song and Xin Yang. Quadratic suffices for over-parametrization via matrix chernoff bound.

In page 6, there is a section talking about ''finite neuron case''. It might be reason to slightly mention about ''infinite neuron case''. The work by Arora et al and Lee et al, shows a connection between infinite case and (ridge) kernel regression.

Sanjeev Arora, Simon S Du, Wei Hu, Zhiyuan Li, Russ R Salakhutdinov, and Ruosong Wang. On exact computation with an infinitely wide neural net. (NeurIPS)

Jason D Lee, Ruoqi Shen, Zhao Song, Mengdi Wang, and Zheng Yu. Generalized leverage score sampling for neural networks. (NeurIPS)

---

> ### Author Response · Authors · 2020-11-17
> **Response to Reviewer1**
>
> Thank you for your encouraging review and the suggested references -- we have included them in the revised submission.

---

### Official Review · AnonReviewer4 · 2020-10-28
**Recommendation to accept on "Deep Equals Shallow for ReLU Networks in Kernel Regimes"**

**Rating:** 7
**Confidence:** 5

**Review:**

The paper shows that the kernel derived from deep fully-connected networks on the sphere have the same approximation properties as their two-layer counterpart for ReLU activations. This implies the limitations of the kernel framework for studying the benefits of such deep networks. The authors derive the asymptotic eigenvalue decay of dot-product kernels from differentiability properties of the kernel function.

Overall, I vote for accepting. The paper is overall well-written and has a solid contribution. It addresses the limitation of the kernel framework, which is very interesting and phenomenal. The experiments are clear to illustrate the theory.

Some concerns are as follows:

The paper only considers ReLU activations and step activations (Section 3.3). Can the analyses be generalized to other network architectures?

The proof only analyzes the kernels on the sphere. It seems hard to generalize the proof to other larger region like R^d. The extension seems to be limited from my side.

Some related literature are missing. To name a few:
Z. Fan and Z. Wang. Spectra of the conjugate kernel and neural tangent kernel for linear-width neural networks (2020).
A. Geifman, A. Yadav, Y. Kasten, M. Galun, D. Jacobs, and R. Basri. On the similarity between the laplace and neural tangent kernels (2020).
Y. Cao, Z. Fang, Y. Wu, D.-X. Zhou, and Q. Gu. Towards understanding the spectral bias of deep learning (2019).
Y. Cho and L. K. Saul. Kernel methods for deep learning (2009).

---

> ### Author Response · Authors · 2020-11-17
> **Response to Reviewer4**
>
> Thank you for the positive review and for pointing out the additional references - we included the missing ones in the revised submission.
>
> - (activations) Indeed, one advantage of our approach is that it is quite generic and can be used as soon as asymptotic expansions of the kernel function can be derived. For instance, it can be applied to positively-homogeneous activations that are powers of the ReLU since expansions of the corresponding kernels easily follow from those of kappa_0 or kappa_1. Another example is smooth activations, for which Theorem 1 implies that the decay is faster than any polynomial, as we discuss in Section 3.3, leading to a much "smaller" RKHS.
>
> - (sphere vs R^d) Note that while general dot-product kernels are difficult to study beyond the sphere, the situation is simpler for the homogeneous case, which arises in particular for ReLU activations. Indeed, as we briefly mentioned in Section 2.1, the RKHS is then precisely described by functions of the form f(x) = |x| g(x/|x|), with g in the RKHS of the kernel with inputs on the sphere. One can further study the spectrum of the covariance operator when the data distribution is spherically symmetric, and it can be shown that the rate of decay is the same as on the sphere (see, e.g., Theorem 5 in Geifman et al, 2020).

---

> > ### Comment · ~Sattar_Vakili1 · 2021-06-25
> > **Endpoint expansios for positively homogeneous activations**
> >
> > Theorem 1 is considered to be applicable to other positively homogeneous activations of the form $\sigma_s(u)=\max(0,u)^s$ using the endpoint expansions which "follow from those of $\kappa_0$ or $\kappa_1$ through integration". Could authors please comment on how this is derived? I guess the resulting eigendecay is in $O(k^{-d-2s+2})$. Is this correct? Thanks.

---

> > > ### Comment · ~Alberto_Bietti1 · 2021-06-25
> > > **reply**
> > >
> > > Hi Sattar,
> > >
> > > This is using the fact that the kernel function for the derivative of an activation is the derivative of the kernel function (see e.g. https://arxiv.org/abs/1602.05897 Lemma 11.c). Then you get $\kappa_s'(u) = s \kappa_{s-1}(u)$, so that you can recover the expansion of $\kappa_s$ from that of $\kappa_{s-1}$ (and by induction, from~$\kappa_0$/$\kappa_1$) by integrating the terms of the expansion. This should yield a decay $k^{-d-2s}$, which matches the decays obtained by Bach (2017). This should have been further clarified, apologies for that.
> > >
> > > Alberto

---

### Official Review · AnonReviewer3 · 2020-10-28
**Similarity between kernels induced by shallow and deep neural networks; Eigenvalue decay**

**Rating:** 6
**Confidence:** 4

**Review:**

-------------- Overview of the paper ----------------

The paper establishes the similarity between kernels (random feature kernel and neural tangent kernel) induced by shallow neural networks and deep ones. In particular, the approximation power is analyzed through the eigenvalue decay of the corresponding kernels --- shallow and deep networks share the same order of decay, expect the difference in parity and constants.

-------------- Contribution and strength ----------------

The message of the paper tends to be a negative perspective on viewing overparameterized neural networks as kernels, which motivates a surge of recent works on showing overparameterized neural networks are easy to optimize and generalize as well as kernels. Yet viewing neural networks as kernels cannot explain the advantage of neural networks, and leaves a large gap between the established theory and practice. This paper, provides theoretical justifications of the insufficiency of the kernel view point.

The organization of the paper is clear and it is relatively easy to read through, although some background on kernels and overparameterized neural networks are needed. Given the tight page limit, the paper makes a good balance between introductory materials and theoretical results.

Theoretically results appear to be correct and sound.

-------------- Weakness ----------------

The data distribution is a bit restricted. However, this is acceptable for later explicit computations of the eigenfunctions and eigenvalues (spherical harmonics and gegenbauer polynomials). Is it possible to consider more general data distributions?

-------------- Questions ----------------

1. The main result (theorem 1) asserts the expansion of the kernel around point 1 and -1. Is it an artifact of the proof (in order to apply integration by part)? Or it in fact reveals some interesting facts on distinguishing identical and orthogonal data points?

2. In Figure 1, we see 2-layer outperforms other models. This is not covered in the proposed theory --- any intuitions on its connection to Theorem 1? Meanwhile, the network width in ReLU random features experiment is smaller than the sample size, which is not the standard overparameterization discussed in lazy training regime. Can the authors elaborate on the choice of the network width?

---

> ### Author Response · Authors · 2020-11-17
> **Response to Reviewer3**
>
> Thank you for the thorough review and positive comments.
>
> - (on the data distribution) Note that the description of the RKHS itself (as given by eq. (9)) is independent of the particular data distribution. Nevertheless, the data distribution does play a role when studying statistical properties of learning algorithms such as ridge regression, in which case the spectrum of the covariance operator comes into play. When the data distribution is uniform on the sphere, this spectrum is the same as the one of the integral operator we study, hence our results also yield tight statistical rates for non-parametric regression, but this is no longer the case for arbitrary distributions (note however that when the distribution has a density w.r.t. the uniform distribution on the sphere one the eigenvalue decays are preserved up to a constant, leading to similar rates).
>
> - (expansions in theorem 1) Note that the expansions are not artifacts of the proof, but assumptions that we leverage in our theorem for obtaining appropriate decays by capturing regularity properties of the kernel function. The fact that these expansions hold for non-integer $\nu$ for common kernels such as those considered in the paper is an intrinsic property of these kernels that allows us to characterize their regularization and approximation properties. We hope this clarifies your concern, please let us know if we can clarify this further.
>
> Regarding Figure 1:
>
> - For the middle figure, the slight differences between different kernels with the same spectral decay (e.g. when varying number of layers) may be due to the difference in constants in the eigenvalues, which may change with depth and with parity (see, e.g., Cor. 2/3).
> - Regarding random features, the 2-layer case seems indeed better than 3-layers. While for 2 layers the expectation of the random feature kernel is the desired dot-product kernel, for 3 layers the expectation is only an approximation of the infinite-width kernel, and our experiment suggests that more neurons are required in order to obtain good approximation compared to the 2-layer case.
> - (scaling of network width) Here we focus simply on random features with only the last layer trained, in order to avoid the complications of the linearization step in NTK/lazy training. We focus on scalings m = sqrt(n) because they are known to be statistically optimal in various regimes (see e.g. Rudi and Rosasco, 2017), and because they are more realistic than the typical NTK scalings where m typically needs to be larger than n, a setting which is not be useful in practice since the computational cost would then be larger than simply using the exact kernel along with the representer theorem. Nevertheless, we also tried m = n and found an even more prominent performance gap between 2 and 3 layers, confirming our observation that 2 layers are preferable for finite-width random ReLU feature models.

---

### Official Review · AnonReviewer2 · 2020-10-28
**Interesting improvement in analyzing expressive power of kernels**

**Rating:** 6
**Confidence:** 3

**Review:**

Summary:

This paper analyzed the expressive power of kernels by studying the reproducing kernel Hilbert space (RKHS) associated with the kernels. Specifically, the authors analyzed the eigenvalue decays in terms of the power series expansions of the kernel function around some points, which is related to the RKHS of the kernel. This analysis can be used to recover some previous results. Besides, using this analysis, the authors have shown several interesting results, including that NTK (which corresponds to fully-connected ReLU networks with infinite width, small learning rate, and proper initialization) with any depth has the same RKHS. The main result also has other corollaries about other kinds of kernels, e.g., Laplace kernel and infinitely differentiable kernels. Experiments were done to validate the theoretical results on synthetic datasets and MNIST/Fashion-MNIST.

Pros:

1. Analyzing the role of depth in deep neural networks is a very important problem, and people observed empirically depth separation phenomena, i.e., deeper networks have more expressive power. The authors analyzed this problem and showed that NTK constrained to the unit sphere does not have the depth separation phenomenon, which makes this paper interesting.

2. The authors did experiments in a nice way to validate the theoretical results. They did experiments on both synthetic datasets and real datasets, compare different kernels and repeat the experiments multiple times.

3. The experimental methodologies and theoretical computations appear to be correct.

Cons:

1. As mentioned by the authors, there is a recent concurrent work (Chen & Xu, 2020) that also obtains the result that NTK with any depth have the same RKHS, but the authors did not provide enough comparisons to that work. It would be better if a more detailed comparison can be provided.

2. The experiments are only done on synthetic datasets/MNIST/Fashion-MNIST with RF/NTK. It would be more convincing if the authors could do experiments with more kernels (e.g., Laplace kernel) or on larger datasets (e.g., CIFAR).

Recommendation:

I tend to vote for accepting this paper. The main reason is that the problem analyzed in this paper is important, and the results in this paper are interesting and kind of surprising. It showed the limitations of NTK in explaining the depth separation phenomenon, which could give people more insights and understandings about NTK.

Supporting arguments for recommendation:

See "Pros", especially point 1 there.

Questions for the authors:

Please address the cons mentioned above.

Additional feedback:

The lines in Figure 1 are a bit hard to distinguish, and it would be better to plot it more clearly, e.g., somehow separate the lines farther away.

---

> ### Author Response · Authors · 2020-11-17
> **Response to Reviewer2**
>
> Thank you for the thorough review and positive comments.
>
> 1. Regarding the concurrent work by Chen&Xu, we did not initially provide a detailed comparison as the paper appeared online only a few days before the submission deadline. We now included a more detailed comparison following the related work section. Note that the approach is quite different: their work shows equality of the NTK RKHS with that of the Laplace kernel by showing a double inclusion through asymptotic decays of Taylor coefficients of the kernel function kappa. In contrast, we obtain precise descriptions of the RKHS and regularization properties of the kernel by studying eigenvalue decays in the basis of spherical harmonics, which in particular implies the equality of the deep NTK and Laplace RKHSs due to the fact that RKHS norms are equivalent when the eigenvalue decays are the same. In Chen&Xu, the asymptotic decay of Taylor coefficients is obtained by characterizing singularities on complex analytic extensions of the kernel function, while we only require asymptotic expansions for non-negative reals t, which are sufficient for obtaining the spectral decay.
>
> 2. Thank your for your suggestions regarding experiments, we have included the Laplace kernel for comparison, which behaves similarly to the NTK, as predicted by the theory. Note that we decided to avoid more complex vision datasets such as Cifar10, since those typically require more structured kernels such as convolutional kernels for better performance, while our theory focuses on the role of depth for fully-connected models and our experiments present simple illustrations of our theory.
>
> 3. Thank you for your comment regarding the curves in Figure 1, we did our best to make them readable. Note that the fact that many of them are hard to distinguish is in agreement with our theory, in particular for different numbers of layers with the same kernel (RF or NTK). Indeed, in our setup with non-smooth f* the theory of ridge regression implies that all kernels considered achieve the optimal rate (which depends on the regularity of f*) when one is allowed to tune the regularization parameter (left figure). The middle figure illustrates that the RF kernels would need a smaller regularization parameter (hence solve a more ill-conditioned optimization problem) compared to NTK/Laplace in order to reach the optimal rate achieved in the left figure, but the behavior is again very similar for different kernels with the same decay.

---

### Author Response · Authors · 2020-11-17
**General response to reviewers**

We thank all reviewers for their encouraging reviews and thoughtful comments. We addressed specific comments in separate responses to each reviewer.

We have uploaded a revised version of the paper with some clarifying modifications based on the reviews, including a more extended comparison to the concurrent work by Chen & Xu (we note that the paper appeared online only a few days before the submission deadline and uses a different approach), as well as additional experiments with the Laplace kernel.

---

### Decision · Program_Chairs · 2021-01-07
**Final Decision**

**Decision:**

Accept (Poster)

**Comment:**

This paper analyzes the expressive power of NTK corresponding to deep neural network. It is shown that the depth hardly affects the behavior of the spectrum of the corresponding integral operator, which indicates that depth separation does not occur as long as NTK is considered.
The analysis is novel and gives a significant insight to the NTK research literature. The theoretical framework considered in this paper is considerably broad and potentially can be applied to several types of activation functions (while only ReLU is analyzed as a concrete example in the paper). Moreover, some numerical experiments are conducted that support the validity of the theoretical analysis.
All reviewers are positive on this paper. I agree with their evaluations. For these reasons, I think this paper is worth acceptance.